

# Comparative analysis of the gut bacteria of the relict gull (*Larus Relictus*) and black-necked grebe (*Podiceps Nigricollis*) in Erdos Relic Gull National Nature Reserve in Inner Mongolia, China

Yaru Zhao, Dulan Bao, Ying Sun, Yajie Meng, Ziteng Li, Rui Liu, Jiwei Lang, Li Liu and Li Gao

Faculty of Biological Science and Technology, Baotou Teachers' College, Baotou, Inner Mongolia, China

## ABSTRACT

The gut microbiota promotes host health by maintaining homeostasis and enhancing digestive efficiency. The gut microflora in wild birds affects host physiological characteristics, nutritional status, and stress response. The relict gull (*Larus Relictus,* a Chinese national first-class protected species) and the black-necked grebe (*Podiceps Nigricollis,* a secondary protected species) bred in the Ordos Relic Gull National Nature Reserve share similar feeding habits and living environments but are distantly related genetically. To explore the composition and differences in the gut microbiota of these two key protected avian species in Erdos Relic Gull National Nature Reserve and provide a basis for their protection, 16S rRNA gene high-throughput sequencing was performed and the gut microbial diversity and composition of the relict gull (*L. Relictus*) and black-necked grebe (*P. Nigricollis*) was characterized. In total, 445 OTUs (operational taxonomic units) were identified and classified into 15 phyla, 22 classes, 64 orders, 126 families, and 249 genera. Alpha diversity analysis indicates that the gut microbial richness of the relict gull is significantly lower than that of the black-necked grebe. Gut microbe composition differs significantly between the two species. The most abundant bacterial phyla in these samples were Proteobacteria, Firmicutes, Fusobacteria, and Bacteroidetes. The prominent phylum in the relict gull was Proteobacteria, whereas the prominent phylum in the black-necked grebe was Firmicutes. The average relative abundance of the 17 genera identified was greater than 1%. The dominant genus in the relict gull was *Escherichia-Shigella*, whereas *Halomonas* was dominant in the black-necked grebe. Microbial functional analyses indicate that environmental factors exert a greater impact on relict gulls than on black-necked grebes. Compared with the relict gull, the black-necked grebe was able to use food more efficiently to accumulate its nutrient requirements, and the gut of the relict gull harbored more pathogenic bacteria, which may be one reason for the decline in the relict gull population, rendering it an endangered species. This analysis of the gut microbial composition of these two wild avian species in the same breeding grounds is of great significance, offers important guidance for the protection of these two birds, especially relict gulls, and provides a basis for understanding the propagation of related diseases.

Corresponding authors
Li Liu, liuli4304842@126.com
Li Gao, gaoli8905@163.com

## INTRODUCTION

Microorganisms are ubiquitous and widely distributed across various environments. Human and animal intestines contain a large number of microorganisms (*Bäckhed et al., 2005*). Dysregulation of the gut microbiome is associated with several diseases in humans and animals (*Dworkin et al., 2006*; *Bäckhed et al., 2012*; *Hsiao et al., 2013*; *Garrett, 2015*). The gut microbiota performs many protective and metabolic functions necessary for host health, including food processing (*Trompette et al., 2014*), digestion of complex polysaccharides that cannot be digested by the host, pathogen replacement, vitamin synthesis (*Neish, 2014*), regulation of insulin sensitivity, fat storage (*Clarke et al., 2012*), and modulating host lipid and glucose metabolism (*Evans, Morris & Marchesi, 2013*; *Wall et al., 2010*).

Avian species occupy significant ecological niches. Changes in the gut microflora of wild birds affect their physiological characteristics, nutritional status, and stress responses (*Laviad-Shitrit et al., 2019*). The gut microorganisms of many rare birds, including Jankowski's bunting (*Emberiza jankowskii*) (*Shang, 2021*) and whooper swans (*Cygnus cygnus*) (*Wang et al., 2021*) have been studied. Relevant research has provided guidance for enacting policies to protect rare species. Wild avian species are known to host emerging human infectious diseases (*Mackenzie & Jeggo, 2013*), and bird migration facilitates the spread of pathogens across multiple geographic areas (*Mackenzie & Jeggo, 2013*). Therefore, studying the gut microorganisms of wild birds can hinder the spread of related pathogenic microorganisms, and studying the intestinal microorganisms of avian species plays an important role in the protection and management of wild birds.

Relict gulls (*Larus relictus*) belong to the Tertiary relict species, family Gullidae, and order Charadriiformes. It is a typical bird species endemic to desert and semi-desert habitats. It is recognized as a vulnerable species by the International Union for the Conservation of Nature (IUCN) and is a national grade I-protected animal in China (*Wang et al., 2013*). Four relatively independent breeding populations of relict gulls exist in the world, and the Ordos population is the main body of the global relict gull population (*Liu et al., 2008*; *Wang et al., 2020a*). One of the most important breeding grounds for this population is T-A Nur in Ordos City, Inner Mongolia Autonomous Region, China, which is an important national nature reserve for relict gulls. They mainly feed on aquatic insects and invertebrates, and their breeding period is from May to July.

Black-necked grebes (*Podiceps Nigricollis*) are medium-sized waterbirds that belong to the Grebe family. They breed in freshwater and saltwater, and congregate in lakes and coasts during winter. They mainly feed on aquatic invertebrates by diving, and occasionally feed on small amounts of aquatic plants. Their breeding period is from May to August. They are rare in China and have been listed as a national second-class protected animal.

During the breeding period, relict gulls and black-necked grebes reach the Ordos Gull Nature Reserve and breed there (*Song et al., 2022*). Relict gulls are the most important protected species, and black-necked grebes are the protected species with the largest breeding population in this area. Food, living environment, and genetic factors are the factors with greatest impact on gut microbes. Relict gull and black-necked grebe breeding in the Ordos Relic Gull Nature Reserve exhibit similar eating habits and living environments. However, these two avian species are very distantly related genetically. To study the gut microbial composition of two avian species with the same living environment but relatively large genetic distance, in June 2021, fecal samples from breeding relict gulls and black-necked grebes in the Ordos Gull Nature Reserve were collected, and the composition of the gut microorganisms of these two avian species were analyzed by 16S rRNA high-throughput sequencing. This study provides a foundation guiding policies for the protection of these two important avian species, especially the protection of the relict gull as a first-class protected animal, and provides insights for combatting the spread of related diseases.

## MATERIALS & METHODS

### Study area and sample collection

Erdos Relic Gull National Nature Reserve (109°14′15″E to 109°23′6″E; 39°42′49N ″ to 39°51′12″N) is located in the middle of Ordos City, Inner Mongolia Autonomous Region, China. It is a major breeding ground for the Ordos population of relict gulls. The temperate continental climate is mainly affected by northwest circulation and polar cold air, with obvious seasonal changes. The vegetation transitions latitudinally from typical grassland to desertified grassland with sparse vegetation and mostly sandy plants. Local animals are mainly wetland birds, typical grassland animals, and reptiles.

Six fecal samples of relict gulls and 5 fecal samples of black-necked grebes were collected in the sampling sites (Fig. 1). To ensure sample collection period consistency, fecal samples were collected in June 2021. Relict gulls and black-necked grebes leave feces around their nests. We approached the birds' nests by following the personnel in the reserve for an in-depth investigation. Samples were collected from each nest. Fresh upper layer fecal samples were placed in 5 mL sterile centrifuge tubes, transported to the laboratory on dry ice, and stored at −80 °C for further assays.

### Ethics statement

This study was conducted in accordance with the requirements for animal care and ethics in China. Non-invasive techniques (*Darimont, Reimchen & Bryan, 2008*) were employed to obtain fecal samples. The animal study was reviewed and approved by the Animal Ethics and Welfare Committee (AEWC) of Baotou Teachers' College. The management authorities of Ordos City in Inner Mongolia agreed to collect relict gull and black-necked grebe fecal samples.

![PeerJ]

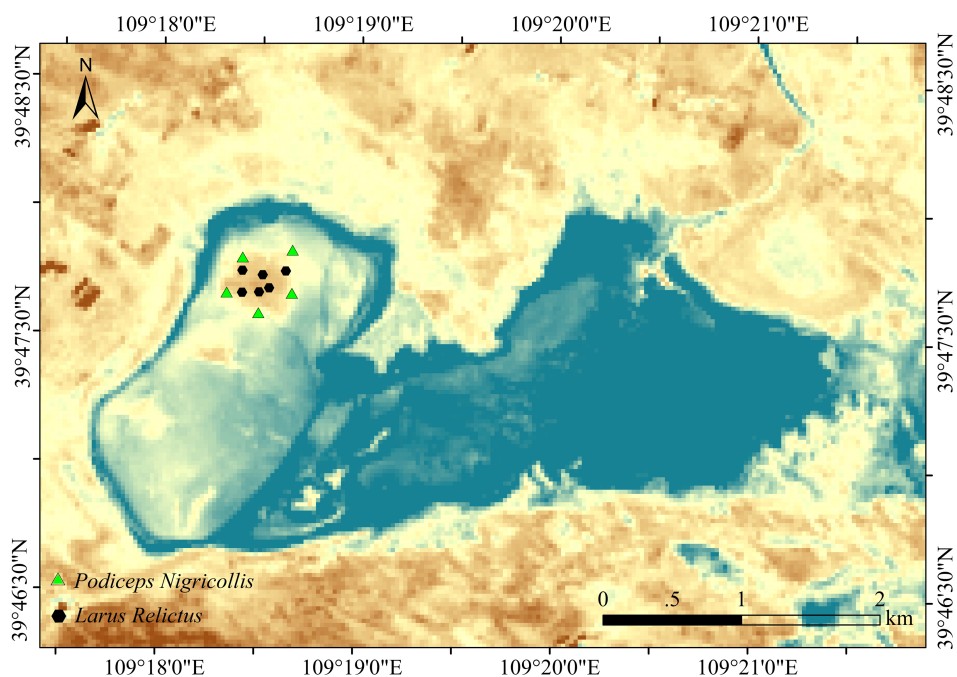

**Figure 1** Study area and sampling point.

## DNA extraction, 16S rRNA high-throughput sequencing, and data analysis

Total DNA was extracted from fecal samples with a QlAamp Fast DNA Stool Mini Kit (Qiagen, Hilden, Germany) according to manufacturer's instructions. The bacterial 16S rRNA gene V3–V4 hypervariable region was amplified and PCR product was assessed by 2% agarose gel electrophoresis. The amplicon was purified, quantified, and sequenced using an Illumina Novaseq 6,000 platform. After sequencing data were filtered for quality control, clean data were clustered into operational taxonomic units (OTU) with 97% sequence identity. The community compositions of the samples were analyzed at various taxonomic levels. Sample alpha diversity indices, including Chao1 and Ace indices, were calculated using the OTU table in QIIME to evaluate the richness and diversity of the bacterial species. Beta diversity analysis was conducted to detect differences in bacterial composition between the relict gull and black-necked grebe. Scatter plots were produced using NMDS and UPGMA according to unweighted and weighted UniFrac distances. Significant microbiota differences between the two avian species were analyzed using (linear discriminant analysis (LDA) Effect Size) (LDA > 4.0). The functions of bacteria with a relative abundance of >1% were predicted using the Kyoto Encyclopedia of Genes and Genomes (KEGG) database. Figures were plotted using R software using a specific method described in a previous study (*Liu et al., 2022*).

## RESULTS

### Statistical analyses of sequence data

A total of 769,048 readable sequences were obtained from 11 samples collected from two avian species, with 51,164 to 78,257 (mean 69,913 $\pm$ 8,415) effective sequences obtained from each sample. Using 97% sequence conservation as cutoff, 445 OTUs were identified, with 241 $\pm$ 34 mean OTU per sample. The OTUs were classified into 15 phyla, 22 classes, 64 orders, 126 families, and 249 genera. The Rarefaction and Shannon index curves plateaued, suggesting that deeper sequencing would have no significant effect on microbial diversity (Figs. 2A, 2B).

A total of 285 UTO were present in relict gulls and black-necked grebes. Sixty OTUs were present in the gut microbiota of the relict gull only, while 100 OTUs were present in the gut microbiota of the black-necked grebe only (Fig. 2C). Thirteen phyla were found in the relict gull and black-necked grebe (Fig. 2D). There were two more gut microbiota phyla in the relict gull, Acidobacteria and Kiritimatiellaeota, which did not exist in black-necked grebe. One hundred and seventy six genera were found in both avian species. Sixteen genera were found in the gut microbiota of relict gulls, and 57 in black-necked grebes (Fig. 2E).

### Alpha diversity and beta diversity analyses

Alpha diversity analysis indicated dramatic differences between relict gull and black-necked grebe were observed in Chao1 (266.43 $\pm$ 38.99 and 317.93 $\pm$ 15.46) and ACE (255.53 $\pm$ 33.79 and 309.21 $\pm$ 12.47) indices ($P < 0.05$) (Figs. 3A, 3B). These results indicate that gut bacteria diversity differed significantly between species.

Beta diversity analysis was conducted to detect differences in bacterial composition between relict gull and black-necked grebe. Scatter plots were produced using NMDS and UPGMA according to unweighted and weighted UniFrac distances. Different fecal samples from the same bird species displayed obvious clustering trends, supporting comparative data showing a dramatic difference in microbial composition between these two bird species (Figs. 4A–4B, 4D, 4E). The ANOSIM test was used to detect significance in differences in $\beta$ diversity. The results revealed that a significant difference in the composition of gut bacteria exists between the two bird species (Figs. 4C, 4F).

### Composition of each species' gut bacterial community

The bacterial composition of each sample was tested. Fifteen phyla were identified. Four of these (Proteobacteria, Firmicutes, Fusobacteria, and Bacteroidetes) were present with a relative abundance of more than 1% (Figs. 5A, 5C). The proportion of these phyla in aggregate was 97.68% for each sample. The most dominant phylum was Proteobacteria in the relict gull, and Firmicutes in the black-necked grebe. A total of 249 bacterial genera were identified, among which 17 had an average relative abundance of more than 1%. These were *Escherichia-Shigella*, *Halomonas*, *Catellicoccus*, *Halolactibacillus*, *Gottschalkia*, *Fusibacter*, *Cetobacterium*, *Vibrio*, *uncultured_bacterium_f_Enterobacteriaceae*, *Lactobacillus*, *Marinobacterium*, *Candidatus_Arthromitus*, *Tissierella*, *Proteiniclasticum*, *Sporosarcina*, *Acetoanaerobium* and *Epulopiscium*. The most dominant genus in the relict

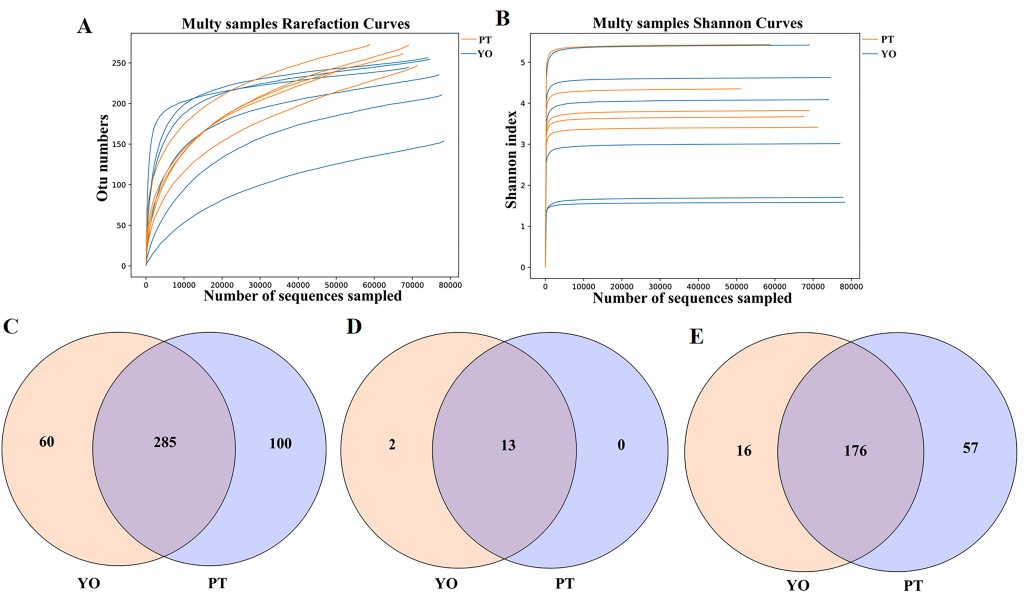

**Figure 2  Feasibility analysis and OTU distribution of the sequencing data.** (A) Bacterial rarefaction curves for the samples; (B) Shannon curves for all the samples; (C) Gut bacterial OTUs distribution in each group; (D) Gut bacterial phylum distribution in each group; (E) Gut bacterial genus distribution in each group. YO, relict gulls; PT, black-necked grebes.

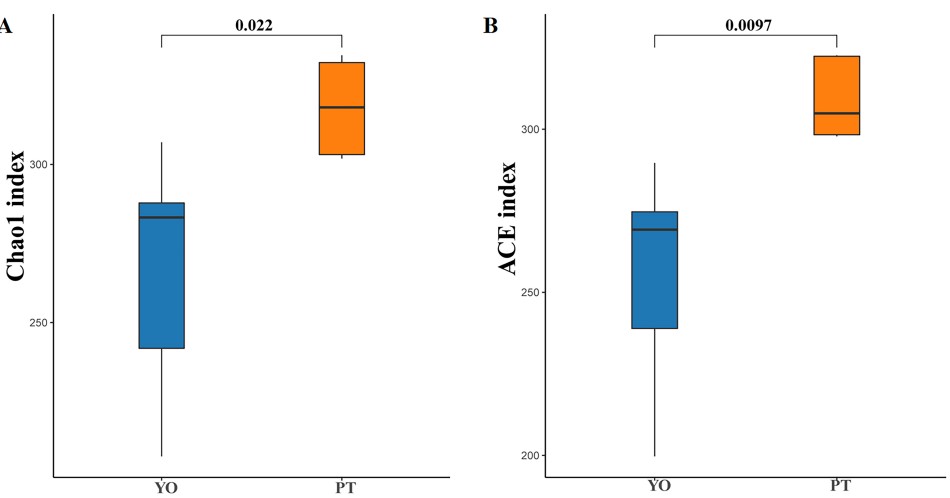

**Figure 3  Gut bacterial α diversities.** (A) Chao1 diversity; (B) ACE index of bacteria in each sample. YO, relict gulls; PT, black-necked grebes.

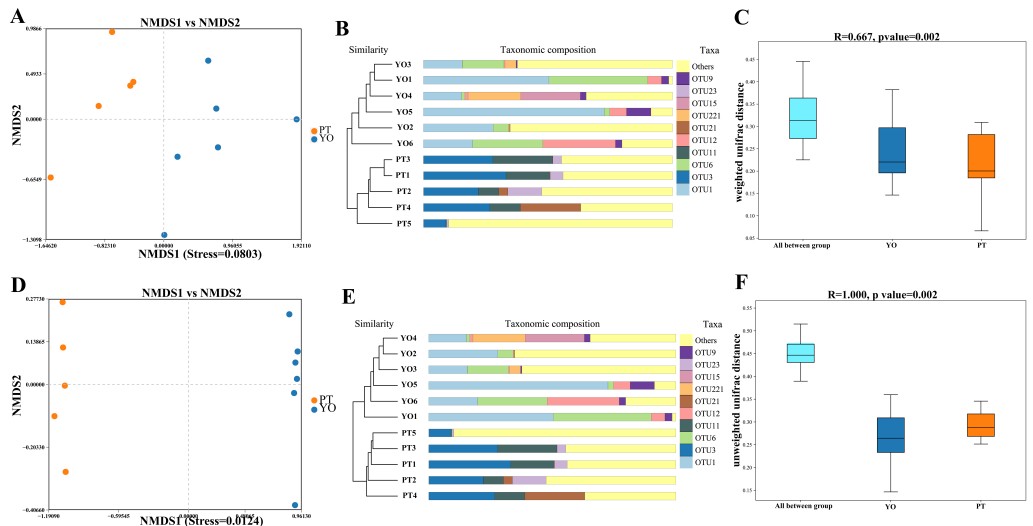

**Figure 4** **Gut bacterial β diversities.** (A) Weighted unifrac distance NMDS plots; (B) Weighted unifrac distance UPGMA analysis; (C) ANOSIM analysis for weighted unifrac. YO, relict gulls; PT, black-necked grebes.

gull was *Escherichia-Shigella*, and in the black-necked grebe it was *Halomonas* (Figs. 5B, 5D).

## Differences between gut microbial compositions of the two bird species

The gut microbiota composition of the relict gull and black-necked grebe were analyzed to identify significant differences. LEfSe analysis results (Fig. 6) indicated that, at the phylum level, the abundance of Fusobacteria was substantially higher in the relict gull than in the black-necked grebe. At the genus level, the abundance of *Escherichia_Shigella*, *Catellicoccus*, *Cetobacterium*, uncultured_bacterium_f_*Enterobacteriaceae*, *Lactobacillus*, *Candidatus_Arthromitus* and *Sporosarcina* was significantly higher in relict gull than in black-necked grebe, while the abundance of *Halomonas*, *Halolactibacillus*, *Fusibacter*, *Gottschalkia*, *Marinobacterium*, *Proteiniclasticum*, *Acetoanaerobium*, *Epulopiscium* and *Tissierella* was significantly higher in the black-necked grebe than in the relict gull (LDA > 4.0, $P < 0.05$).

## Bacterial community functional prediction

Bacterial functions with a relative abundance >1% were predicted using Picrust2. In the Class 1 level, the main functions of these bacteria were Metabolism, Environmental information processing, Genetic information processing, Human diseases, Cellular processes, and Organism systems. Among all functions, the relative abundance of Human diseases and Environmental information processing was significantly higher in relict gulls than in black-necked grebes, while the relative abundance of Cellular processes was significantly lower in relict gulls than in black-necked grebes (Table 1). At the Class 2 level, the main functions of these bacteria were Global and overview mapping, Carbohydrate metabolism, Amino acid metabolism, Membrane transport, Energy metabolism, and

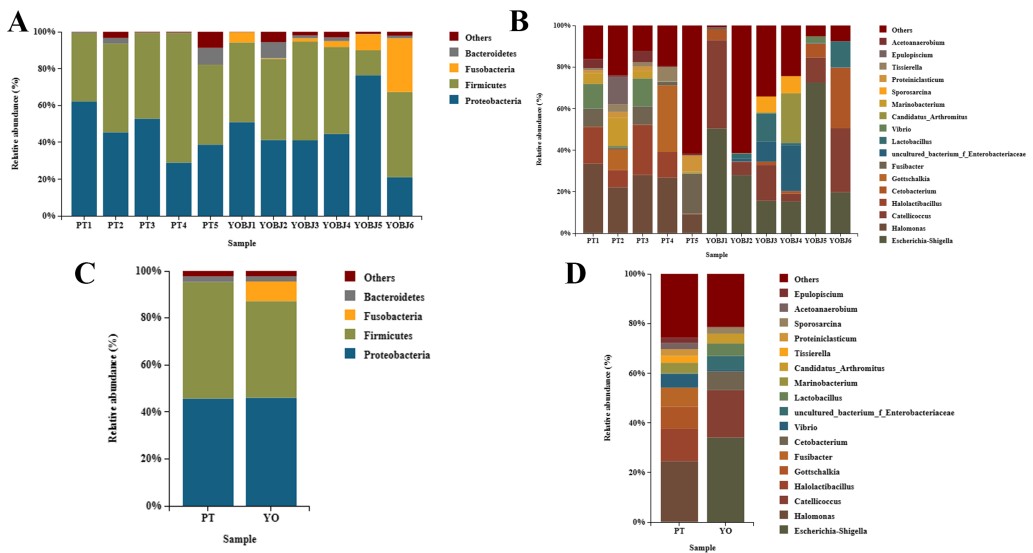

**Figure 5** **Bar graph of the bacterial relative abundance at phylum (A, C) and genus (B, D) level.** Bacteria with relative abundance (%) of more than 1% were revealed. Others, bacterial with a relative abundance of under 1%. YO (YOBJ), relict gulls; PT, black-necked grebes.

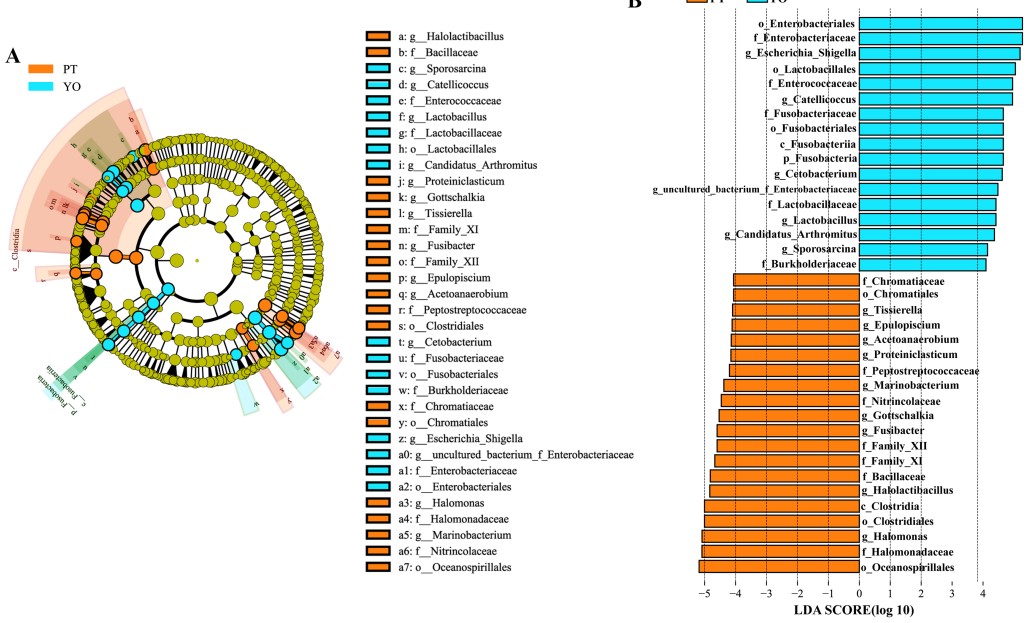

**Figure 6** **Linear discriminant analysis effect size (LEfSe) analysis of the microbial species with dramatic differences between relict gulls and black-necked grebes.** (A) Cladogram. The classification at the level of phylum, class, order, family and genus were showed from the inside to the outside. (B) Plot from LEfSe analysis. LDA score > 4.0, *P* < 0.05. YO, relict gulls; PT, black-necked grebes.

Metabolism of cofactors and vitamins. Among these functions, the relative abundance of Carbohydrate metabolism, Membrane transport, Glycan biosynthesis and metabolism, Metabolism of terpenoids and polyketides, Metabolism of other amino acids, and Lipid metabolism were significantly higher in relict gull than in black-necked grebe, while the relative abundance of Cell motility, Metabolism of cofactors and vitamins, Amino acid metabolism, and Global and overview maps were significantly lower in relict gull than in black-necked grebe (Table 1). At the Class 3 level, the main functions of these bacteria were Metabolic pathways, Secondary metabolite biosynthesis, Antibiotic biosynthesis, Microbial metabolism in diverse environments, and ABC transporters. Among these functions, the relative abundances of Amino sugar and nucleotide sugar metabolism, and ABC transporters were significantly higher in relict gulls than in black-necked grebes, whereas the relative abundances of Amino acid and Secondary metabolite biosynthesis were significantly lower in relict gulls than in black-necked grebes (Table 1).

## DISCUSSION

Avian species occupy a very important ecological niche. Thus, the gut microorganisms of many rare birds have received attention (*Shang, 2021*; *Wang et al., 2021*). Relevant research has provided guidance for the protection of rare species. Wild avian species are known to host emerging human infectious diseases (*Mackenzie & Jeggo, 2013*), and bird migration facilitates the spread of pathogens across multiple geographic areas (*Mackenzie & Jeggo, 2013*). Therefore, the study of the gut microorganisms of wild avian species can not only help us understand their life history and related mechanisms, but also hinder the spread of bird related pathogenic microorganisms. Therefore, the study of intestinal microorganisms of avian species plays an important role in the protection and management of wild avian species.

In the current study, 16S rRNA high-throughput sequencing was conducted to compare the gut microbial diversity and composition of relict gulls and black-necked grebes bred in the Erdos Relic Gull National Nature Reserve. The results revealed a dramatic difference in the $\alpha$ diversity of the gut microorganisms of these two avian species. The gut microbial richness of the black-necked grebe was significantly higher than that of relict gull. Because these two avian species breed in the same area, differences in gut microbial richness may be mainly caused by genetic differences between these species. These results indicate that genetic factors exert a greater impact on gut microorganisms than short-term survival environments and food, consistent with the results of another study (*Wang, 2022*), that found significant differences in gut microbial diversity among five crane species sharing the same feeding conditions. Another previous study found a negative correlation between gut microbial diversity and pathogenic bacteria in the intestine (*Xiang et al., 2019*), indicating that relict gulls may harbor more pathogenic bacteria in their intestines.

Significant differences in gut microbial composition were observed between relict gulls and black-necked grebes. The most abundant bacterial phyla in these two avian species were Proteobacteria, Firmicutes, Fusobacteria, and Bacteroidetes, consistent with previous studies on gut microbes in wild birds (*Jandhyala et al., 2015*; *Wang et al., 2020b*). The most

| Function | YO | PT | |
|---|---|---|---|
| **Table 1** **Bacterial community functional prediction.** | | | |
| **Function** | **YO** | **PT** | |
| Class 1 | Relative abundance | Relative abundance | *P*-values |
| Cellular processes | 3.08 ± 0.45 | 4.42 ± 0.22 | 0.000527 |
| Human diseases | 3.18 ± 0.24 | 2.65 ± 0.14 | 0.003039 |
| Environmental information Processing | 8.41 ± 0.56 | 7.58 ± 0.44 | 0.034119 |
| Metabolism | 76.05 ± 0.61 | 76.46 ± 0.39 | 0.262796 |
| Organismal systems | 1.35 ± 0.06 | 1.30 ± 0.08 | 0.361889 |
| Genetic information processing | 7.92 ± 0.74 | 7.59 ± 0.58 | 0.478947 |
| Class 2 | Relative abundance | Relative abundance | *p*-values |
| Carbohydrate metabolism | 9.87 ± 0.32 | 8.67 ± 0.19 | 9.35E−05 |
| Cell motility | 0.79 ± 0.36 | 2.02 ± 0.13 | 0.000266 |
| Membrane transport | 5.40 ± 0.47 | 4.36 ± 0.25 | 0.002654 |
| Glycan biosynthesis and metabolism | 1.21 ± 0.09 | 1.02 ± 0.04 | 0.004635 |
| Metabolism of cofactors and vitamins | 3.87 ± 0.15 | 4.29 ± 0.17 | 0.004736 |
| Amino acid metabolism | 6.15 ± 0.58 | 7.27 ± 0.17 | 0.0062055 |
| Metabolism of terpenoids and polyketides | 1.17 ± 0.06 | 1.05 ± 0.04 | 0.010424 |
| Metabolism of other amino acids | 1.55 ± 0.06 | 1.43 ± 0.05 | 0.010833 |
| Global and overview maps | 40.06 ± 0.57 | 40.91 ± 0.35 | 0.024044 |
| Lipid metabolism | 2.25 ± 0.11 | 2.11 ± 0.09 | 0.071736 |
| Nucleotide metabolism | 3.96 ± 0.37 | 3.65 ± 0.25 | 0.174007 |
| Replication and repair | 2.92 ± 0.27 | 2.74 ± 0.21 | 0.300703 |
| Signal transduction | 2.97 ± 0.32 | 3.18 ± 0.28 | 0.317638 |
| Cellular community - prokaryotes | 1.62 ± 0.09 | 1.69 ± 0.10 | 0.34544 |
| Xenobiotics biodegradation and metabolism | 1.26 ± 0.10 | 1.35 ± 0.22 | 0.516895 |
| Translation | 3.29 ± 0.39 | 3.17 ± 0.30 | 0.613705 |
| Folding, sorting and degradation | 1.54 ± 0.06 | 1.53 ± 0.06 | 0.721742 |
| Energy metabolism | 3.90 ± 0.09 | 3.91 ± 0.22 | 0.973499 |
| Class3_name | Relative abundance | Relative abundance | *p*-values |
| Amino sugar and nucleotide sugar metabolism | 1.24 ± 0.13 | 0.94 ± 0.05 | 0.002088 |
| Biosynthesis of amino acids | 3.23 ± 0.19 | 3.61 ± 0.14 | 0.007099 |
| ABC transporters | 3.88 ± 0.24 | 3.44 ± 0.18 | 0.01129 |
| Biosynthesis of secondary metabolites | 7.14 ± 0.15 | 7.38 ± 0.13 | 0.02677 |
| Glycolysis/Gluconeogenesis | 1.13 ± 0.13 | 0.99 ± 0.05 | 0.065509 |
| Purine metabolism | 2.22 ± 0.20 | 2.04 ± 0.11 | 0.117888 |
| Microbial metabolism in diverse environments | 4.40 ± 0.11 | 4.24 ± 0.22 | 0.222044 |
| Pyrimidine metabolism | 1.73 ± 0.18 | 1.61 ± 0.13 | 0.262597 |
| Pyruvate metabolism | 1.16 ± 0.07 | 1.13 ± 0.02 | 0.284799 |
| Two-component system | 2.53 ± 0.31 | 2.74 ± 0.27 | 0.316706 |
| Metabolic pathways | 16.08 ± 0.23 | 16.21 ± 0.22 | 0.428072 |

**Table 1** (*continued*)

| Function | YO | PT | |
|---|---|---|---|
| Ribosome | 2.14 ± 0.26 | 2.07 ± 0.22 | 0.634384 |
| Biosynthesis of antibiotics | 5.23 ± 0.11 | 5.25 ± 0.08 | 0.856164 |
| Carbon metabolism | 2.59 ± 0.02 | 2.60 ± 0.09 | 0.859003 |
| Quorum sensing | 1.51 ± 0.11 | 1.50 ± 0.04 | 0.950863 |

**Notes.**

YO, Relict gull; PT, Black-necked grebe.

abundant bacterial phylum in the relict gull was Proteobacteria, whereas the dominant phylum in black-necked grebes was Firmicutes. Studies have shown that Firmicutes is more abundant in phytovorous animals, and its main function is to disintegrate cellulose into volatile fatty acids that can be absorbed by the host, improve the nutrient usage rate, regulate T cells to enhance host immunity, prevent intestinal inflammation, and maintain the ecological balance of gut microorganisms (*Fernando et al., 2010*; *Guan et al., 2017*), Proteobacteria are mainly composed of some pathogenic bacteria, which is an indicator of gut flora instability (*Shin, Whon & Bae, 2015*). Previous studies have shown that, in humans, increased Firmicutes/Bacteroidetes (F/B) ratios are correlated with obesity (*Ley et al., 2006*; *Turnbaugh et al., 2009*). In our study, the F/B ratio of black-necked grebes was significantly higher than that of relict gulls (123.25 *vs* 96.23), indicating that black-necked grebes can use food resources more efficiently to maintain body health, which may also be one reason for efficient reproduction among its population. The ability of the relict gull to use food resources is relatively poor, and more potential pathogenic bacteria may be present in their intestines, resulting in it being an endangered species.

Two gut microbiota phyla existed only in the relict gull: *Acidobacteria* and *Kiritimatiellaeota*. *Acidobacteria* are a widespread bacterial phylum in natural environments including extreme environments (*Lee, Ka & Cho, 2008*). *Kiritimatiellaeota* is a bacterial phylum that regulates arginine and fatty acid synthesis. It often exists in the guts of high-altitude animals (*Guo et al., 2021*) and facilitates high-altitude animals' use of low-fat foods to supply energy to adapt to extreme environments. These two bacterial phyla only existed in the gut of relict gulls and were not found in the gut of black-necked grebes, indirectly indicating that environmental factors exert a greater impact on relict gulls, in accord with previous studies showing that the living environment needs of relict gulls were quite harsh (*Zhang et al., 1993*).

At the genus level, 17 bacterial genera had an average relative abundance of >1%. The dominant genus in the relict gull was *Escherichia-Shigella*, whereas *Halomonas* was dominant in black-necked grebes. *Escherichia-Shigella* can prompt the body to initiate an inflammatory state (*Soares et al., 2012*), and its adhesion to host tissues, underlying invasive chronic *Escherichia* infection may lead to persistent peripheral inflammation (*Small et al., 2013*). *Halomonas* has also been found to be a human pathogenic bacterium (*Stevens et al., 2009*), suggesting that some microorganisms in the intestinal tracts of these avian species in this area are unfavorable to their survival, and relevant agencies should monitor these microbes and take relevant protective measures to prevent their spread.

Relict gulls displayed significantly higher abundance of environmentally derived microorganisms such as *Uncultured bacterium f Enterobacteriaceae* (*Osaili et al., 2018*), *Cetobacterium* (*Ramírez et al., 2018*), and *Candidatus_Arthromitus* (*Del-Pozo et al., 2010*) in their gut than did black-necked grebes. These results indicate that environmental factors exert a greater effect on relict gull gut microbes than on black-necked grebe gut microbes.

Gut microorganism functions in these two avian species were further analyzed using the KEGG database. Their main Class 1 level functions were related to Metabolism, and the proportion of microbe functions associated with Human diseases and Environmental information processing was significantly higher in relict gulls than in black-necked grebes, indicating that the relict gull carries more pathogenic bacteria and is more susceptible to environmental influences. At Class 2 level, Carbohydrate metabolism, Membrane transport, Glycan biosynthesis and metabolism, Metabolism of terpenoids and polyketides, Metabolism of other amino acids, and Lipid metabolism were significantly higher in relict gull than in black-necked grebes, while the relative abundance of Cell motility, Metabolism of cofactors and vitamins, Amino acid metabolism and Global and overview maps were significantly lower in relict gulls than in black-necked grebes, indicating that different avian species had different metabolic flora in their intestines, which may be related to differences in feeding behaviors between the two species. At Class 3 level, Amino sugar and nucleotide sugar metabolism and ABC transporters were significantly higher in relict gulls than in black-necked grebes, while the relative abundance of Biosynthesis of amino acids and Biosynthesis of secondary metabolites were significantly lower in relict gulls than in black-necked grebes, indicating that the relict gull had strong metabolic capacity and poor nutrient synthesis ability, so they cannot make good use of food and environmental resources for their own functions, which may be one reason underlying their smaller population.

These differences in gut microorganisms may lead to reduced host functions in the relict gull, negatively impacting its resilience and population, rendering it an endangered species. In future protection efforts, more attention should be paid to changes in gut microorganisms when estimating relict gull population health, and to reducing pathogenic microorganism numbers in their environment. We also found that some gut microorganisms in relict gulls were obtained from their food. It is thus possible to determine the main food types based on gut microorganisms and provide foods in their habitats that will positively impact their gut microbiota, thereby achieving the goal of protecting this rare species. This study reveals significant implications for the analysis of the gut microbial composition of these two wild avian species in the same breeding grounds, which offers important guidance for the protection of these two species, especially the relict gull as a first-class protected animal, and provides a basis for minimizing the spread of related diseases.

## CONCLUSIONS

Summing up, in the present study, high-throughput sequencing was used to analyze fecal samples of relict gulls and black-necked grebes breeding in Erdos Relic Gull National Nature

Reserve in Inner Mongolia, China. The results showed that dramatic differences existed in the gut microbial diversity and composition of relict gulls and black-necked grebes and that environmental factors impacted relict gulls more heavily than black-necked grebes. Compared with the relict gull, the black-necked grebe was able to use food more efficiently to accumulate its own nutrients, and there were more pathogenic bacteria in the gut of the relict gull, which may be one reason for the decline in the relict gull population, rendering it an endangered species. This analysis of the gut microbial composition of these two wild avian species in the same breeding grounds is of great significance, and offers important guidance for the protection of these two bird species, especially relict gulls, and provides insights into the propagation of related diseases.

### Funding

This research was funded by Impact of Habitat Change on Relict Gulls in Ordos Relict Gull National Nature Reserve and Wetland Restoration Effectiveness Assessment Project, grant number 30437020; the High-level Talents Introduced Scientific Research Startup Fund Project of Baotou Teachers' College, grant number BTTCRCQD2020-003; the Baotou Teachers College High-level Research Achievement Cultivation Project, grant number BSYKJ2021-ZQ03. The funders had no role in study design, data collection and analysis, decision to publish, or preparation of the manuscript.

### Grant Disclosures

The following grant information was disclosed by the authors:
Impact of Habitat Change on Relict Gulls in Ordos Relict Gull National Nature Reserve and Wetland Restoration Effectiveness Assessment Project: 30437020.
High-level Talents Introduced Scientific Research Startup Fund Project of Baotou Teachers' College: BTTCRCQD2020-003.
Baotou Teachers College High-level Research Achievement Cultivation Project: BSYKJ2021-ZQ03.

### Competing Interests

The authors declare there are no competing interests.

### Author Contributions

- Yaru Zhao performed the experiments, analyzed the data, prepared figures and/or tables, authored or reviewed drafts of the article, and approved the final draft.
- Dulan Bao performed the experiments, prepared figures and/or tables, and approved the final draft.
- Ying Sun performed the experiments, prepared figures and/or tables, and approved the final draft.
- Yajie Meng performed the experiments, prepared figures and/or tables, and approved the final draft.

 

- Ziteng Li performed the experiments, authored or reviewed drafts of the article, and approved the final draft.
- Rui Liu performed the experiments, prepared figures and/or tables, and approved the final draft.
- Jiwei Lang performed the experiments, prepared figures and/or tables, and approved the final draft.
- Li Liu conceived and designed the experiments, prepared figures and/or tables, and approved the final draft.
- Li Gao conceived and designed the experiments, prepared figures and/or tables, authored or reviewed drafts of the article, and approved the final draft.

## Animal Ethics

The following information was supplied relating to ethical approvals (i.e., approving body and any reference numbers):

The Animal Ethics and Welfare Committee of Baotou Teachers College approved the study.

## Field Study Permissions

The following information was supplied relating to field study approvals (i.e., approving body and any reference numbers):

The management authorities of Ordos City in Inner Mongolia agreed the collecting of relict gull and black-necked grebe fecal samples.

## DNA Deposition

The following information was supplied regarding the deposition of DNA sequences:

The datasets are available at NCBI: PRJNA788023 (SRX13401783, SRX13401784, SRX13401785, SRX13401786, SRX13401791, SRX13401792), PRJNA956598, SRR24263419–SRR24263422.

## Data Availability

The datasets presented in this study can be found in online repositories. The names of the repository/repositories and accession number(s) can be found below: http://www.ncbi.nlm.nih.gov/bioproject/ PRJNA788023 and http://www.ncbi.nlm.nih.gov/bioproject/PRJNA956598.

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
