# Peer review of "Comparative analysis of the gut bacteria of the relict gull (Larus Relictus) and black-necked grebe (Podiceps Nigricollis) in Erdos Relic Gull National Nature Reserve in Inner Mongolia, China"

_PeerJ, doi:10.7717/peerj.15462_

## Round 0.1 · original submission · Major Revisions

Dear Authors,

Thanks for submitting your work to PeerJ.

As you can see from the reviewers' comments, a major revision decision has been recommended for your study. Please go through the comments and provide your point-by-point responses when submitting your revised manuscript to the journal for further consideration.

We look forward to receiving your revision soon.

Kind regards,
Liang

Reviewer 1 ·

Basic reporting

Abstract section:
Line 20, “Relict Gull (YO)”. What does YO mean? Latin names in italics should be written in brackets. Relict Gull or Relict gull, please keep the uniform style throughout your manuscript.
Line 20, “Black-necked Grebe (PT)”. What does PT mean? Latin names in italics should be written in brackets. Black-necked Grebe or Black-necked grebe, please keep the uniform style throughout your manuscript.
Line 22, “were gained, they were classifified”. Please revise this sentence and the wrong word.
Line 27, All bacterial phyla need to be corrected to italic. Please keep the uniform style throughout your manuscript.
Introduction section:
Line 45, “desert-semi desert habitats”. Should be desert and semi-desert habitats.
Lines 45-46, “a vulnerable species by the World Conservation Union”. You mean IUCN?
Line 49, “Ordos population is the main body of the global relict gull”. How many?
Lines 60-61, “It feeds mainly by diving, The food was mainly aquatic invertebrates, and occasionally a small amount of aquatic plants.”. Please revise this sentence.
Line 87, “relict gull (L. Relictus)”. There is no need to write Latin names. Please check the full text and correct such errors.
Lines 91-93, “Studies had shown that gut microbes have important links with food and environment, so do the gut microbes of relict gulls and black-necked grebes reproducing under the same environmental conditions tend to be similar?”. I don't know what you mean by this sentence.
Materials & Methods:
Line 108, “nest of the birds and picked up into”. The nests of these two birds should be on the water, right? How do people approach? Will there be droppings from young birds? How to ensure the freshness of feces?
Line 121, “was clustered into the Operational taxonomic unit (OUT) according”. OUT should be OTU. Please check through the manuscript to revise OUT.

The description of statistical methods and the description of chart making methods are missing.
Results section:
Lines 176-184, Phyla and genera need to be written in italics.
Discussion section:
Line 247, “Wang et al., 2020a”. Incorrect format. Because there is no Wang et al., 2020b. Please check your references. I also found another error in 2020b, please modify it.

The gut microbiomes of birds are affected by many factors, such as genetic factors. Genetic factors were not discussed.

Experimental design

no comment

Validity of the findings

no comment

Reviewer 2 ·

Basic reporting

Authors should provide more backgroung information on this study. The English is poor for understanding.

Experimental design

Authors should add more information on the study site and sample collection.

Validity of the findings

There was no direct evidence to support this conclusion.

Additional comments

This paper provide some valuable information on the gut bacteria of relict gull and black-necked grebe. However, their results are descriptive, and authors did not provide direct evidence to support the results. For example, what is the differences in diet and environment for the two species? These factors are important in determining the differences in gut bacteria of the two species. Secondly, why did authors compare the gut bacteria of the two bird species which are not close-relate species? The authors did not give more detailed background information on this study. Thirdly, the English is poor for understanding. Detailed comments see below:
L44-100, the logic of INTRODUCT is very confused, and need to be rewritten. Authors need to provide more background information on the ecology of two species. Specially, authors should note why they studied the two bird species which are not close-relate species.
L103-105, authors should provide more information on the study site, including climate, vegetation et al. I suggest that authors can add a figure of the study site.
L105-107, How to select fecal sample? Fresh sample? One sample representing one individual?
L167, 173, how to define the “dominate phylum”, “dominant genus”?
L207-226, this part should be moved to INTRODUCTION.
L227-236, these conclusions were not supported by direct evidences. Specially there was no data on the diet of black-necked grebes.
L240-242, the prediction of the low gut microbial richness relating to the less adaptable to the environment is far-fetched. As authors mentioned, one of most important breeding grounds of relict gull population distribute in this study site, which indicating that the population adapt the current environments. Smaller population may be related to other factors, for example human hunting.

Reviewer 3 ·

Basic reporting

no comment

Experimental design

no comment

Validity of the findings

no comment

Additional comments

This study analyzed the fecal microbes from wild Relict Gull (Larus Relictus) and Black-necked Grebe (Podiceps Nigricollis). The research on gut microbiota of wild animals is an interesting topic and is of great significance on further understanding the certain species, protection of endangered animals and even the coevolution between commensal bacteria and host. According to this manuscript, these two avian species share the same habitat and have similar feeding habits, while there are differences in bacterial communities between them. The methods and results are sufficient, however, the authors do not clearly describe their aims and significances of the study. Therefore, I think this manuscript needs corresponding modifications mainly on the parts of Abstract, Introduction and Discussions before accepted.
1. In the section of Abstract, the sentence “Gut microbiota promotes host health by maintaining homeostasis and improving digestion efficiency.” (Line 18-19) is not enough as the background and aims of this study.
2. The structure of the Introduction is chaotic, and there is also a lack of detailed research progress on gut microbiome of avian species.
3. The habitat mentioned in this manuscript (Erdos Relic Gull National Nature Reserve, Line 102-103), where the authors collected the samples, should be provided with clear location information, such as longitude and latitude.
4. “A total of 6 relict gulls’ fecal samples and 5 black-necked grebes’ fecal samples were collected.” (Line 105-106) Please describe in detail whether these samples were from different individuals in different nests. Besides, when collecting fecal samples, how to ensure that the samples are of the tested species? Is there any possibility of confusion?
5. “These results indicating that there was no difference in the diversity of gut bacteria between the two species of birds, but there was a significant difference in the richness of the gut bacteria.” (Line 152-154) All the four indices (Shannon, Simpson, Chao1 and ACE) can reflect the alpha diversity, but the emphasis on evenness and richness is different.
6. The Discussion and Conclusion sections excessively repeated the results. They should systematically and emphatically discuss potential causes and effects of the microbiota differences between the two avian species that lives in the same area.
7. There are many grammatical and lexical errors in the manuscript. The language should be modified by fluent English experts.

---

## Round 0.2 · accepted · Accept

After going through the revised manuscript, I agree that the authors have addressed all of the reviewers' comments. I am happy with the current version and the manuscript is ready for publication.

Reviewer 3 ·

Basic reporting

no comment

Experimental design

no comment

Validity of the findings

no comment

Additional comments

The authors have fully addressed the comments and revised the manuscript. I think this revised manuscript can be accepted by Peer J.